# Unsupervised Learning of Efficient and Robust Speech Representations

## Abstract

We present an unsupervised method for learning speech representations based on a bidirectional contrastive predictive coding that implicitly discovers phonetic structure from large-scale corpora of unlabelled raw audio signals. The representations, which we learn from up to 8000 hours of publicly accessible speech data, are evaluated by looking at their impact on the behaviour of supervised speech recognition systems. First, across a variety of datasets, we find that the features learned from the largest and most diverse pretraining dataset result in significant improvements over standard audio features as well as over features learned from smaller amounts of pretraining data. Second, they significantly improve sample efficiency in low-data scenarios. Finally, the features confer significant robustness advantages to the resulting recognition systems: we see significant improvements in out-of-domain transfer relative to baseline feature sets, and the features likewise provide improvements in four different low-resource African language datasets.

## 1 Introduction

The representation of the input to machine learning models strongly determines the difficulty faced by the learning algorithm, how much data the learner will require to find a good solution, and whether it will generalize out of sample and out of the domain of the training data. Representations (or features) that encode relevant information about data enable models to achieve good performance on downstream tasks, while representations that are invariant to factors that are not relevant to downstream tasks can further improve generalization and sample efficiency. Traditionally, many invariances were hard-coded in feature extraction methods. For example in image representations, geometric and photometric invariance has been investigated (Mundy et al., 1992; Van De Weijer et al., 2005). For acoustic representations, it is known that the standard MFCC features are sensitive to additive noise and many modifications have been proposed to overcome those limitations (Dev & Bansal, 2010; Kumar et al., 2011).

In this paper, we investigate the potential of unsupervised representation learning for speech from large-scale, unlabelled and noisy data using bidirectional contrastive predictive coding (§2–§3). In previous work, van den Oord et al. (2018) showed that the unidirectional variant of this objective results in representations that are useful for frame-level phoneme classification (even with simple linear classifiers), but in this paper, we go further: we vastly increase the scale of the pretraining data to be larger and much noisier and more diverse, and we also evaluate the learned representations in a speech recognition (ASR) system, rather than on the less interpretable per-frame phone prediction.

In our experiments (§4), we find that our acoustic representations improve ASR system performance relative to standard spectrogram and log-filterbank features, and also relative to features pretrained from smaller corpora that are more matched to the training/test domain. A previous attempt at integrating pretrained acoustic representations into an ASR system from Schneider et al. (2019) found that they were able to outperform mel-frequency features in both sample efficiency and overall performance, but their work used only clean, read English speech (from the LibriSpeech dataset) as pretraining data, and evaluated the features when used to recognize speech in a rather similar domain (read Wall Street Journal text). They found that increases in the amount of pretraining data led, in some cases, to diminished performance. However, we find the opposite: our best pretrained features are those learned from the largest and noisiest speech data, with performance far surpassing those obtained by pretraining only on the relatively clean LibriSpeech.

In addition to the standard ASR evaluation where the systems are evaluated on the same domain as they are trained on, we look at two further aspects of our features: their sample efficiency and whether their use confers robustness to domain shifts in the resulting system. Here again, we find compelling results. First, for a smaller speech recognition task (using a simpler convolutional architecture than is common), an ASR model trained with our representations only needs 10% of the labelled data to achieve the same result as a the same model trained on spectrogram features using 100% of the training data. Second, we evaluate the robustness of our representations by estimating how invariant they are to domain and language shifts. To do so, an ASR model is trained using our representations on one dataset but evaluated on the test sets of other datasets. In this experiment, we find that the representations derived from the large pretraining dataset lead the ASR model to be much more robust to domain shifts, compared to both log filterbank features as well as to pretraining just on LibriSpeech. Finally, we also evaluate the representations on four low-resource African languages (i.e. Amharic, Fongbe, Swahili, Wolof) and show that our representations significantly outperform both standard features and those pretrained only on clean English data.

In summary, we confirm several increasingly common patterns that may be discerned in the literature on unsupervised representation learning, across a variety of modalities. First, scale matters: good representation learning requires a large amount of data. Second, contextualized representations (that are sensitive to both past and future), are valuable. And finally, unsupervised representations consistently improve sample efficiency, performance, and robustness of downstream tasks.

## 2 CONTRASTIVE PREDICTIVE CODING: CPC

Unsupervised representation learning methods rely on differentiable objectives which quantify the degree to which representations have succeeded at capturing the relevant characteristics in data. Mutual information measures relationships between random variables (Fano & Hawkins, 1961). Mutual information maximization techniques, that learn representations that describe data by maximizing mutual information between data and representation variables, have been explored for a long time in unsupervised representation learning (Linsker, 1988; Bell & Sejnowski, 1995). However, since the exact computation of mutual information is not tractable for continuous variables, many estimators have been proposed for enabling unsupervised representation learning with neural networks (Belghazi et al., 2018; van den Oord et al., 2018; Hjelm et al., 2019).

Contrastive predictive coding (van den Oord et al., 2018, CPC) is a mutual information maximization method that has been successfully applied to many modalities such as images and speech (Hénaff et al., 2019; Bachman et al., 2019; Schneider et al., 2019). The objective is designed to extract features that allow the model to make long-term predictions about future observations. This is done by maximizing the mutual information of these features with those extracted from future timesteps. The intuition is that the representations capture different levels of structure dependent on how far ahead the model predicts. For example, if the model only predicts a few steps ahead, the resulting representations can capture local structures. On the other hand, if the model predicts further in the future, the representations will need to infer "slow features" (Wiskott & Sejnowski, 2002); more global structures such as phonemes, words and utterances in speech.

The overall unsupervised learning process is visualized in Figure. 1. Given a raw audio signal of length $L$ ($\boldsymbol{x} = x_1, x_2, \ldots, x_L$, $x_i \in \mathbb{R}$ where $x_i$ represents the acoustic amplitude at time $i$), a function $g_{enc}$ encodes the audio signals into vector representations ($\boldsymbol{z} = \boldsymbol{z}_1, \boldsymbol{z}_2 \ldots, \boldsymbol{z}_M$, $\boldsymbol{z} \in \mathbb{R}^{d_z}$). Next, an auto-regressive function $g_{ar}$, such as a recurrent neural network, summarizes the past representations and produces context vectors ($\boldsymbol{c} = \boldsymbol{c}_1, \boldsymbol{c}_2 \ldots, \boldsymbol{c}_M$, $\boldsymbol{c} \in \mathbb{R}^{d_c}$). The representations are learned to maximize mutual information between context vectors ($\boldsymbol{c}$) and future latent representations ($\boldsymbol{z}$) as follows:

$$\sum_{t,k} I(\boldsymbol{c}_t, \boldsymbol{z}_{t+k}) = \sum_{t,k} \sum_{\boldsymbol{c}_t, \boldsymbol{z}_{t+k}} p(\boldsymbol{c}_t, \boldsymbol{z}_{t+k} \mid k) \log \frac{p(\boldsymbol{z}_{t+k} \mid \boldsymbol{c}_t, k)}{p(\boldsymbol{z}_{t+k})}.$$

Since the mutual information is not tractable for high dimensional data, it is common to use a lower-bound on the mutual information such as InfoNCE (van den Oord et al., 2018) which is a loss function based on noise contrastive estimation (Gutmann & Hyvärinen, 2010). Given a set $Z = \{\boldsymbol{z}_1, \ldots \boldsymbol{z}_N\}$ which contains one positive sample from $p(\boldsymbol{z}_{t+k}|\boldsymbol{c}_t)$ and $N - 1$ negative samples

from a "noise" distribution $p(\boldsymbol{z})$, the approximated lower-bound is written as

$$I(\boldsymbol{c}_t, \boldsymbol{z}_{t+k}) \geq \mathbb{E}_Z \left[ \log \frac{f_k(\boldsymbol{c}_t, \boldsymbol{z}_{t+k})}{\frac{1}{N} \sum_{\tilde{\boldsymbol{z}} \in Z} f_k(\boldsymbol{c}_t, \tilde{\boldsymbol{z}})} \right] = \mathcal{L}_{tk}^{NCE}, \tag{1}$$

where $f_k(\boldsymbol{c}_t, \boldsymbol{z}_{t+k})$ is a scoring function. We used the standard log-bilinear model as follows:

$$f_k(\boldsymbol{c}_t, \boldsymbol{z}_{t+k}) = \exp(\boldsymbol{c}_t^T \boldsymbol{W}_k \boldsymbol{z}_{t+k}).$$

The loss function we maximize is a sum of the InfoNCE loss for each step, $\mathcal{L}^{NCE} = \sum_t \sum_k \mathcal{L}_{tk}^{NCE}$ and the negatives are uniformly sampled from representations in the same audio signal ($\boldsymbol{z}$) and/or mini-batch.

## 3 METHODS

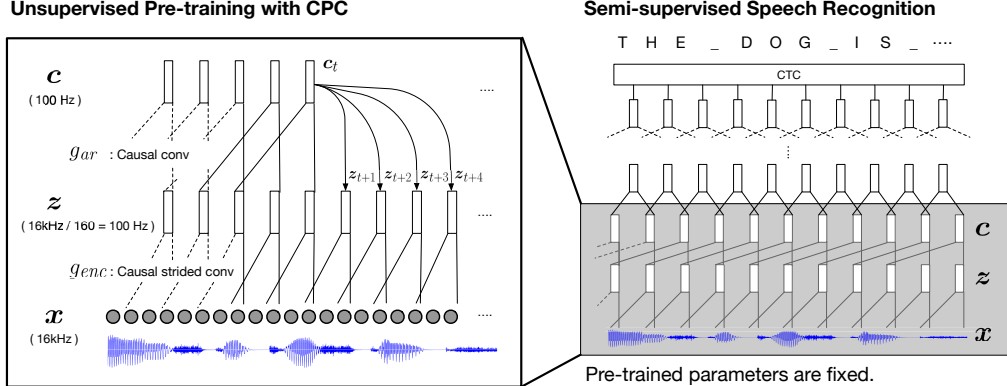

Figure 1: **Left**, unsupervised representation learning with forward contrastive predictive coding. The learned representations are fixed and used as inputs to a speech recognition model (**Right**).

In this section, we describe our models and objectives for unsupervised representation learning and downstream speech recognition. First, an acoustic feature extractor is trained with a bidirectional variant of contrastive predictive coding (CPC) on an unlabeled audio dataset. Next, the parameters of this model are frozen and its output representations are used as input to train various speech recognition models, potentially on a different or smaller labeled dataset (Figure 1).

### 3.1 UNSUPERVISED LEARNING WITH BI-DIRECTIONAL CONTRASTIVE PREDICTIVE CODING

Following the success of bidirectional models in representation learning (Peters et al., 2018; Devlin et al., 2019), we extend the original CPC method explained above with bidirectional context networks. The encoder function $g_{enc}$ is shared for both directions but there are two auto-regressive models ($g_{ar}^{fwd}$ and $g_{ar}^{bwd}$) which read encoded observations ($\boldsymbol{z}$) from the forward and backward contexts, respectively. The forward and backward context representations $\boldsymbol{c}_t^{fwd}, \boldsymbol{c}_t^{bwd}$ are learned with separate InfoNCE losses. When they are used for downstream tasks, a concatenation of two representations $\boldsymbol{c_t} = [\boldsymbol{c}_t^{fwd}; \boldsymbol{c}_t^{bwd}]$ is used. A similar technique has been used in image representation learning where representations are learned along different spatial dimensions (Hénaff et al., 2019).

All audio signals have a sampling rate of 16kHz and we normalize the mean and variance of the input signals over each utterance in order to mitigate volume differences between samples. For architectures, we use encoder and auto-regressive models similar to Schneider et al. (2019). The encoder function $g_{enc}$, is a stack of causal convolutions with kernel sizes (10, 8, 4, 4, 4, 1, 1) and stride sizes (5, 4, 2, 2, 2, 1, 1), corresponding to a receptive field of 10 ms of audio. For auto-regressive functions, we use a 13 layer causal convolution architecture with kernel sizes (1, 2, ..., 12, 13) and stride size 1, for both forward and backward functions. Layer-normalization across the temporal and

feature dimensions is applied to every layer. Also, each layer has dense skip connections with layers below as in DenseNet (Huang et al., 2017). The objective function we optimize is the sum of the forward and backward InfoNCE losses (eq.1).

## 3.2 SEMI-SUPERVISED SPEECH RECOGNITION

Once the acoustic representations are trained, the resulting context vectors ($c$) are used as inputs to character-level speech recognition models which predict transcriptions of audio-signals character by character. The model first predicts frame-level character probabilities with a series of convolution layers while the CTC forward algorithm (Graves et al., 2006) calculates conditional probabilities of a transcription given an audio signal. The model parameters are trained to maximize likelihood of the data. The training is terminated when the word error rate on development set stops improving or the model is trained more than certain epochs. The models are evaluated on standard word error rate on test data. During training, the parameters in speech recognition models are trained with supervision but the parameters of the pretrained models are fixed. For decoding, we use greedy CTC decoding without a language model (LM) in order to focus on discerning the effects of the acoustic representations, although we also include results with a 4-gram LM to facilitate comparisons with published results.

Common practice in unsupervised representation learning is to evaluate learned representations using a linear classifier. However, we found a simple linear layer followed by a CTC decoder does not have enough capacity to recognize speech. Thus, for our first set of experiments we use a smaller version of DeepSpeech2 (Amodei et al., 2016) to predict the frame-level character probabilities. The model has two 2d-convolutions with kernel sizes (11, 41) and (11, 21) and stride sizes (2, 2) and (1, 2) and one recurrent neural network (GRU) on top of the output from the convolution layers. A linear transformation and a softmax function are applied to predict frame-level character probabilities. We refer to **DeepSpeech2 small** for the model specifics (Amodei et al., 2016). In order to further investigate how the representations interact with larger speech recognition models, we use the time-delay neural networks (**TDNN**) that are commonly used in speech recognition (Collobert et al., 2016; Kuchaiev et al., 2018). These consist of 17 layers of 1d-convolutions followed by 2 fully connected layers. Refer to OpenSeq2Seq for a detailed description.[1] These large models have enough capacity to learn to recognize speech from log-filterbank features with purely supervised learning on sufficient data, so they represent a challenging test case for learned representations.

## 4 EXPERIMENTS AND RESULTS

### 4.1 DATASET

We collected nine publicly available speech datasets which cover a variety of types of speech (e.g. read and spoken), noise conditions and languages. For unsupervised pretraining we used a combination of datasets, removing speech transcriptions for datasets that included those. For semi-supervised learning on top of the reprensenations we used the transcribed datasets following their standard train-test splits. A full overview of all datasets and their statistics can be found in Appendix A.

Since we will ultimately compare ASR systems trained on one dataset but evaluated on the test set of another, we must ensure that the representation of the transcription is consistent. To do so, we use the format of the LibriSpeech dataset, which also ensure our results are comparable with standard speech recognition systems (Kuchaiev et al., 2018). For the other datasets, the transcriptions are lower-cased and unpronounced symbols are removed. We also removed examples that contain numbers as there are multiple ways for transcribing them.

**Large-scale unlabeled speech** We collected large-scale unlabelled speech data from two existing datasets, a subset of Audio Set (Gemmeke et al., 2017) and the audio part of AVSpeech (Ephrat et al., 2018). Among many annotated acoustic events in Audio Set, we only used examples labeled as "speech." We also included the Common Voice (CV)[2] dataset in all the 29 languages available. Note

---

[1] https://nvidia.github.io/OpenSeq2Seq/html/speech-recognition/wave2letter.html
[2] https://voice.mozilla.org

that there are many examples with background noise (e.g. music, conversations etc.) and a variety of languages. We used an opportunistic strategy to gather our 8000h pretraining dataset, and it is conceivable that more careful curation of this dataset can be used to manipulate the properties of the learned representations, but we leave such considerations for future work.

**Read English** In speech recognition research, it is common to train models on read English corpora such as LibriSpeech (Panayotov et al., 2015), the Wall Street Journal (Paul & Baker, 1992) and TIMIT (Garofolo, 1993). We use standard train / test splits for speech recognition evaluation. We also use the Speech Accent Archive (Weinberger & Kunath, 2009, SSA).

**Spoken English** We also evaluate on conversational speech and public speaking datasets. Switchboard (Godfrey et al., 1992) is a standard conversational speech recognition dataset consisting of two-sided telephone conversations. Since the data was recorded more than 10 years ago, it is relatively noisy compared to recent English corpora. Tedlium-3 (Hernandez et al., 2018) is a large spoken English dataset which contains 450 hours of speech extracted from TED conference talks. The recordings are clear but there are some environmental noise (reverberation).

**Low-resource languages** We use speech recognition datasets in four African languages collected in the ALFFA project[3]: Amharic (Tachbelie et al., 2014), Fongbe (A. A Laleye et al., 2016), Swahili (Gelas et al., 2012), Wolof (Gauthier et al., 2016) for evaluation. These languages have unique phonetic properties (e.g. height harmony) and phonemic inventories, making them a good contrast to English. These languages are low-resource, each with 20 hours or less of transcribed speech.

## 4.2 UNSUPERVISED REPRESENTATION LEARNING

We train the model described above (§3.1) with a combination of the large-scale unlabelled data and for the speech recognition part we use the training portions of the transcribed speech recognition datasets (§4.1). Similarly to Schneider et al. (2019)), audio signals are randomly cropped with a window size 149,600 observations (9.35 seconds) and encoded with the model. The bidirectional contrastive predictive coding objective (e.q.,1) with prediction steps ($k$) 12 and negatives ($N$) 10 is optimized with the Adam optimizer with learning rate 0.0001. A batch size of 128 is used as well as a polynomial learning rate scheduler with power 2 and gradient clipping with maximum norm 5.0. Training was terminated at 4.2 million steps based on speech recognition performance on the development set of the LibriSpeech corpus.

## 4.3 SAMPLE EFFICIENCY

We evaluate the pretrained representations by looking at sample complexity on downstream speech recognition tasks. 10% and 100% portions of the Wall Street Journal and LibriSpeech dataset are used to train speech recognition models. We use small and large speech recognition models (**DeepSpeech2 small**, **TDNN**) to see how much information is contained in the representations. The speech recognition models are trained in the similar way as heavily tuned state-of-the-art models (Collobert et al., 2016; Kuchaiev et al., 2018). We refer to Appendix B for a detailed description of training configurations.

Table 1 shows results with **DeepSpeech2 small** models trained on different sizes of the Wall Street Journal corpus. Since the original DeepSpeech2 (Amodei et al., 2016) model uses spectrogram features, we use the same features to train our baselines. **CPC-LibriSpeech** and **CPC-8k** indicate representations are learned from LibriSpeech and 8000h of speech datasets listed above respectively. The ablation experiments show the effectiveness of bidirectional models. The bidirectional models obtain consistent improvements, and unsupervised pretraining with the larger uncurated noisy dataset also shows large improvements. We would like to highlight the comparison with a model trained on spectrogram features with 100% of training data and a model trained on learned representations with 10% of the training data. Interestingly, when a model is small (i.e. models need to rely more on the features since they are capable of less complex transformations of the signal), the learned representations can be as much as ten times more sample efficient compared to spectrogram features.

---

[3] http://alffa.imag.fr

Tables 2 and 3 summarize the results with **TDNN** models trained on different sizes of the Wall Street Journal and LibriSpeech corpora. Following the well-tuned open source models (Collobert et al., 2016; Kuchaiev et al., 2018), our baseline is trained using log-filterbank features. Even if the speech recognition models have a large number of parameters and are trained on plenty of supervised data, the learned representations still provide significant improvements. The pattern continues to hold if we use beam search decoding with a language model.[4] Our **+ LM decoding** results are comparable to the OpenSeq2Seq benchmark, since we used the exact same LM and decoding algorithm as they used (Kuchaiev et al., 2018).

| | WSJ-dev 93 | | WSJ-test 92 | | WSJ-test 93 | |
|---|---|---|---|---|---|---|
| | 10% | 100% | 10% | 100% | 10% | 100% |
| **WSJ** | | | | | | |
| Spectrogram | 80.61 | 54.04 | 77.63 | 48.17 | 78.82 | 51.78 |
| CPC-LibriSpeech (-backward) | 62.05 | 42.54 | 55.13 | 34.82 | 59.44 | 40.14 |
| CPC-LibriSpeech | 55.59 | 38.15 | 47.76 | 30.25 | 54.05 | 35.72 |
| CPC-8k (-backward) | 59.97 | 41.03 | 52.81 | 32.13 | 58.33 | 39.69 |
| CPC-8k | **52.08** | **34.30** | **42.92** | **27.42** | **49.99** | **34.61** |

Table 1: Sample efficiency experiments with **DeepSpeech2 small** trained and evaluated on the **Wall Street Journal**. The numbers are word error rate on development and evaluation sets and the second row indicates the amount of training data used to train each model.

| | WSJ-dev 93 | | WSJ-test 92 | | WSJ-test 93 | |
|---|---|---|---|---|---|---|
| | 10% | 100% | 10% | 100% | 10% | 100% |
| **WSJ** | | | | | | |
| LogFilterbank | 53.63 | 20.72 | 46.77 | 16.78 | 52.28 | 23.26 |
| CPC-LibriSpeech | 37.89 | 15.28 | 32.11 | 11.97 | 40.32 | 15.64 |
| CPC-8k | **36.00** | **14.48** | **29.25** | **10.77** | **37.78** | **14.99** |

Table 2: Sample efficiency experiments with the **TDNN** on the **Wall Street Journal**. The metrics are word error rate on the development and evaluation sets and the second row indicates the amount of training data.

| | LibriSpeech | | | | | | | |
|---|---|---|---|---|---|---|---|---|
| | dev-clean | | dev-other | | test-clean | | test-other | |
| | 10% | 100% | 10% | 100% | 10% | 100% | 10% | 100% |
| **LibriSpeech** | | | | | | | | |
| LogFilterbank (OpenSeq2Seq) | - | 6.67 | - | 18.67 | - | 6.58 | - | 19.61 |
| LogFilterbank (ours) | 19.83 | 6.63 | 38.97 | 18.77 | 19.65 | 6.43 | 41.26 | 20.16 |
| CPC-LibriSpeech | 15.07 | 6.70 | 33.55 | 19.77 | 14.96 | 6.91 | 36.05 | 21.60 |
| CPC-8k | **13.92** | **6.20** | **30.85** | **17.93** | **13.69** | **6.25** | **32.81** | **19.10** |
| **+ LM decoding** | | | | | | | | |
| LogFilterbank (OpenSeq2Seq) | - | 4.75 | - | 13.87 | - | 4.94 | - | 15.06 |
| LogFilterbank (ours) | 12.49 | 4.87 | 28.71 | 14.14 | 12.29 | 5.04 | 31.03 | 15.25 |
| CPC-LibriSpeech | 9.66 | 4.87 | 24.72 | 14.34 | 9.41 | 5.05 | 26.77 | 16.06 |
| CPC-8k | **8.86** | **4.35** | **22.10** | **12.96** | **8.70** | **4.72** | **24.15** | **14.47** |

Table 3: Sample efficiency experiments with the **TDNN** trained and evaluated on **LibriSpeech**. The results are word error rate on the LibriSpeech development and evaluation sets. 10% vs. 100% indicates the amount of training data used. The section in **+ LM decoding** contain results with beamsearch decoding with a 4-gram language model. The underlined (OpenSeq2Seq) scores are taken from public benchmarks.[6]

---

[4] http://www.openslr.org/resources/11/4-gram.arpa.gz

## 4.4 ROBUSTNESS

Robustness to shifts in domain, recording conditions, and the intonation of speech is an important desideratum for a good ASR system, and we hypothesized that the diversity of our largest pretraining regime would improve robustness along these dimensions. In contrast, standard MFCC features have been tested in terms of noise robustness and it is known that such representations are sensitive to additive noise (Zhao & Wang, 2013). Moreover, speech recognition systems developed on top of such features are not robust when they are evaluated on out-of-domain datasets (Amodei et al., 2016).

To test whether our pretraining approach improves robustness, we evaluate speech recognition models trained on the learned representations on many different datasets so as to investigate benefit of using the representations learned from large-scale data. We compare ASR systems on all of the Wall Street Journal and LibriSpeech corpora with the same optimization as explained above and evaluate word error rate on different evaluation sets, such as phone call conversations (Switchboard).

Table 4 summarizes the results on models trained on the Wall Street Journal, LibriSpeech or the Tedlium corpora and evaluated on different evaluation sets. The features trained on large-scale data consistently outperform other representations across different evaluation sets. The speech recognition models trained on the Wall Street Journal perform badly on phone call data in general. However, the representations learned on large datasets are more robust than the representations trained only on read English data (LibriSpeech).

| | WSJ | | LibriSpeech | | Tedlium | | Switchboard |
|---|---|---|---|---|---|---|---|
| | test92 | test93 | test-clean | test-other | dev | test | eval2000 |
| **WSJ** | | | | | | | |
| LogFilterbank | 16.78 | 23.26 | 46.27 | 73.27 | 58.61 | 62.55 | 96.44 |
| CPC-LibriSpeech | 11.89 | 15.66 | 31.05 | 56.31 | 45.42 | 47.79 | 83.08 |
| CPC-8k | **10.77** | **14.99** | **29.18** | **51.29** | **38.46** | **39.54** | **69.13** |
| **LibriSpeech** | | | | | | | |
| LogFilterbank | 14.42 | 21.08 | 6.43 | 20.16 | 26.9 | 25.94 | 61.56 |
| CPC-LibriSpeech | 14.28 | 20.74 | 6.91 | 21.6 | 26.53 | 27.14 | 63.69 |
| CPC-8k | **13.31** | **18.88** | **6.25** | **19.10** | **21.56** | **21.77** | **53.02** |
| **Tedlium** | | | | | | | |
| LogFilterbank | 20.35 | 27.23 | 24.05 | 47.27 | 18.75 | 19.31 | 74.55 |
| CPC-LibriSpeech | 15.01 | 19.52 | 17.77 | 36.7 | 15.28 | 15.87 | 61.94 |
| CPC-8k | **13.17** | **17.75** | **16.03** | **32.35** | **13.67** | **13.88** | **47.69** |

Table 4: Domain transfer experiments to test the robustness of the representations to domain shifts. The models are trained on the **Wall Street Journal**, **LibriSpeech** or **Tedlium** and evaluated on different evaluation sets.

## 4.5 LOW-RESOURCE LANGUAGE

Thus far, all our experiments have compared our representations in terms of their impacts on English recognition tasks (although we know that the pretraining dataset contains samples from many languages). We now turn to the question of whether these representations are suitable for driving recognition different languages with substantially different phonetic properties than English has. Specifically, we look at the performance on four languages—Amharic, Fongbe, Swahili, and Wolof—which manifest a variety of interesting phonological properties that are quite different from English. Evaluating on such languages will provide insights into the phonetic space learned in the representations[7]. Moreover, our non-English languages are low-resource in terms of speech recognition data, but have 2–20 million native speakers each. It is therefore valuable if the representations learned from large-scale unlabelled data can improve low-resource speech recognition. Although there is a small chance that the large-scale pretraining dataset may contain some examples from those languages, we did not add any extra data specifically to improve representations for those languages.

---

[7]The universality of these representations is important since spectrograms, whatever their limitations, are language agnostic.

To test the cross-linguistic value of these features, we trained speech recognition models on low-resource languages (§4.1) and compare how much difference we obtain from using standard features and the learned representations. As these are very small datasets, we trained **DeepSpeech2 small** models with the Adam optimizer with a fixed learning rate of 0.0002 and gradient clipping with maximum norm 25.0. Note that we did not tune architectures and learning methods for particular configurations so as to keep the comparisons fair.

Table 5 summarizes results. Again, we find that the representations trained on large-scale data outperforms other features by a large margin and that the models trained on the representations trained on (English-only) LibriSpeech do not perform as well as standard features. This suggests that the representations learned on large-scale data capture a phonetic space that generalizes across different languages.

| | Amharic | | Fongbe | | Swahili | | Wolof | |
| --- | --- | --- | --- | --- | --- | --- | --- | --- |
| | dev | test | dev | test | dev | test | dev | test |
| Spectrogram | 76.30 | 78.85 | 53.48 | 65.34 | 80.21 | 77.18 | 60.72 | 69.93 |
| CPC-LibriSpeech | 86.53 | 89.01 | 67.08 | 74.09 | 86.8 | 85.51 | 69.31 | 77.47 |
| CPC-8k | **63.51** | **66.10** | **42.50** | **57.20** | **68.07** | **69.23** | **44.04** | **55.41** |

Table 5: Speech recognition on low-resource languages. Models are trained and evaluated on each language based on different features.

## 5 RELATED WORK

Unsupervised learning of representations was an active area of research in deep neural networks. Representations learned with deep Boltzmann machines and auto-encoders trained to reconstruct inputs had initial successes in image and speech recognition (Hinton et al., 2006; Bengio et al., 2007; Vincent et al., 2010; Hinton et al., 2012). After a period of focusing on supervised techniques, unsupervised representation learning has recently seen a resurgence in a variety of modalities (Doersch & Zisserman, 2017; van den Oord et al., 2018; Donahue & Simonyan, 2019; Bachman et al., 2019) and has led to improved results, especially in low-data regimes (Hénaff et al., 2019; Schneider et al., 2019). In natural language processing, pretrained representations can outperform state-of-the-art system even in high data regimes (Mikolov et al., 2013; Devlin et al., 2019).

There have been considerable work on unsupervised speech representation learning. The most frequent aim is to learn representations that correspond to phonetic structure. In order to measure the quality of learned structures many evaluations have been proposed. The Zerospeech challenges explicitly look at correlations between learned representations and phonetic structures (Dunbar et al., 2019). van den Oord et al. (2018) used a frame-level phoneme classification task with a linear classifier to investigate how well the representations can discriminate phonemes. Although the task is easy to evaluate, the results do not correspond to existing downstream tasks. Schneider et al. (2019) applied learned representations to speech recognition and showed the effectiveness of learned representations. However, as we discussed in the paper, many important aspects such as invariances to domain shift and language shift were not carefully assessed.

## 6 CONCLUSION

We presented an unsupervised speech representation learning method that discovers acoustic representations from up to 8000 hours of publicly accessible speech data. We have shown, for the first time, that such pretrained representations lead speech recognition systems to be sample efficient and more robust to domain shifts compared to standard acoustic represenations, and compared to representations trained on smaller and more domain-narrow pretraining datasets. The representations are evaluated on a standard speech recognition setup where the models are trained and evaluated on in-domain data and also on a transfer tasks where the models are evaluated on out-of-domain data. We obtained consistent improvements on different English corpora as well as four low-resource African languages. This suggests we are making progress toward models that implicitly discovers phonetic structure from large-scale unlabelled audio signals.

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

## A   DATASET STATISTICS

Table 6 summarizes dataset statistics.

| Name | Language | Type | Transcript | Avg. utterance length | Total Hours |
|------|----------|------|------------|-----------------------|-------------|
| Audio Set | Multilingual | Not specified | No | 9.9s | 2500h |
| AVSpeech | Multilingual | Not specified | No | 6.4s | 3100h |
| Common Voice | Multilingual | read | Yes | 4.7s | 430h |
| LibriSpeech | English | read | Yes | 12.1s | 960h |
| WSJ | English | read | Yes | 7.8s | 80h |
| TIMIT | English | read | Yes | 3.1s | 5h |
| SSA | English | read | Yes | 16.2s | <1h |
| Tedlium | English | spoken | Yes | 6.0s | 440h |
| Switchboard | English | spoken | Yes | 4.5s | 310h |
| ALFFA | Amharic | read | Yes | 6.7s | 20h |
| ALFFA | Fongbe | read | Yes | 2.9s | 7h |
| ALFFA | Swahili | read | Yes | 3.5s | 11h |
| ALFFA | Wolof | read | Yes | 4.3s | 18h |

Table 6: Summary of Dataset Statistics. All the datasets except for ALFFA dataset are used for unsupervised representation learning.

## B   TRAINING CONFIGURATIONS

For all the speech recognition models, model parameters are initialized with a scaled uniform distribution (Glorot & Bengio, 2010). We used different optimization methods for each model. For the small model, the Adam optimizer with learning rate 0.0002 and gradient clipping with a maximum norm 25.0 is used. For a large model on Wall Street Journal dataset, we used vanilla stochastic gradient descent with learning rate 5.6 and gradient clipping with maximum norm 0.05 is used. For a large model on LibriSpeech dataset, we used Adam optimizer with learning rate 0.0002 and gradient clipping with a maximum norm 5.0. We noticed it is important to tune the learning rate during training for the large model. The polynomial learning rate decay method with power 2.0 is used over 200 epochs. Those training methods are similar to heavily tuned state-of-the-art models (Collobert et al., 2016; Kuchaiev et al., 2018).

## C    STATE-OF-THE-ART SPEECH RECOGNITION SYSTEM

We focused on learning representations and evaluating them in terms of sample efficiency and robustness. Thus, we used TDNN model, which is most commonly used speech recognition architecture, with greedy decoding from CTC. However, many models and decoding strategies have been proposed in speech recognition research. Here, we clarify how our results are comparable with existing results.

Our result with TDNN model on LibriSpeech dataset with learned representation is comparable with published benchmark. As Table 3 summarize, our our CPC-8k feature consistently outperform state-of-the-art results with the same model. The pattern holds even if we use beamsearch decoding with 4-gram language model provided by the open-sourced tool.

Recently, Li et al. (2019) proposed a new model, which is a larger version of TDNN networks, and obtained 3.61 with greedy decoding. They improved the result to 2.78 with 6-gram language model and to 2.58 with additional re-ranking with transformer-XL language model. This is the current state-of-the-art results on LibriSpeech dataset. There are lot of rooms for improvements in terms of speech recognition performance. However, our features are evaluated on a common architecture with decoding method comparable with open-sourced tools and the improvements are consistent across many evaluations.

