# OpenReview forum: "Unsupervised Learning of Efficient and Robust Speech Representations"
_ICLR.cc/2020/Conference — Reject_

### Official Review · AnonReviewer3 · 2019-10-09
**Official Blind Review #3**

**Rating:** 6

**Review:**


Overview:

This work uses contrastive predictive coding (CPC) to learn unsupervised speech representations on large amounts of unlabelled speech data and then uses the resulting features in downstream speech recognition systems. Unlabelled data is obtained from several sources (spanning different languages). Supervised systems are then built on top of these features and sample-efficiency and cross-domain robustness is investigated using English data sets. Finally, the approach is applied to four African languages.

Strengths:

Firstly, the paper is very clearly written and motivated. Secondly, a very relevant problem is tackled in a systematic way; compared to transcribed resources, unlabelled resources are much easier to collect and more widely available. This paper shows that these unlabelled resources can be of great benefit in downstream tasks and on languages where few resources are available. Thirdly, the experiments are carried out very systematically to support the claims of the paper: that bidirectional CPC-based feature learning improves same efficiency (they show that much less labelled data is required to achieve the same performance as when using more substantial labelled data with conventional features), and that it improves robustness to out-of-domain data. They perform these experiments on both English and truly low-resource languages.

Weaknesses:

There are two main weaknesses to the paper. Firstly, as the authors note themselves, unsupervised CPC-based speech feature learning was developed and considered in previous work, and has also been subsequently investigated by others. The main technical contribution is therefore only in changing the unidirectional architecture to bidirectional. Secondly, the paper does a very poor job of linking this work with previous work. The work in [1] is very related. In Section 5, the ZeroSpeech challenges are mentioned briefly (with a single citation), but over the last decade there has been substantial work in this community specifically looking at exactly the main problem addressed in this paper (unsupervised speech representation learning). It would be of great benefit to situate this work within that context, and I would recommend that the paper at least mention [2] to [9].

Overall assessment:

Although technical novelty is limited (first weakness), I think there is novelty in the paper's systematic experimental investigation, including ASR experiments on truly low-resource languages. The conclusions of this work also has practical implications for the ASR community. The second weakness can be addressed by amending Section 5. I therefore assign a "Weak Accept" to the paper.

Questions, suggestions, typos, grammar and style:

- In Figure 1, it might be useful to indicate the autoregressive nature of the context vectors by adding arrows in-between the $c$ blocks on the top left. (In the text it says an RNN is used.)
- p.7: "... are suitable for driving recognition different languages ...". A typo or grammatically incorrect sentence.
- p. 9: "Tts without t" -> "TTS without T"
- p. 9: "african" -> "African" (check all citations for capitalization)

Missing references:

1. https://arxiv.org/abs/1904.03240
2. A. Jansen et al. A summary of the 2012 JHU CLSP workshop on zero resource speech technologies and models of early language acquisition. ICASSP, 2013.
3. Badino, L., Canevari, C., Fadiga, L., & Metta, G. (2014). An auto-encoder based approach to unsupervised learning of subword units. in ICASSP.
4. Versteegh, M., Anguera, X., Jansen, A. & Dupoux, E. (2016). The Zero Resource Speech Challenge 2015: Proposed Approaches and Results. In SLTU-2016 Procedia Computer Science, 81, (pp 67-72).
5. Renshaw, D et al. (2015). A Comparison of Neural Network Methods for Unsupervised Representation Learning on the Zero Resource Speech Challenge. Interspeech.
6. R. Thiolliere et al. A  hybrid  dynamic  time  warping-deep  neural  network  architecture  forunsupervised acoustic modeling. Interspeech. 2015
7. https://arxiv.org/abs/1811.08284
8. https://arxiv.org/abs/1702.01360
9. https://arxiv.org/abs/1709.07902


**Experience Assessment:**

I have published in this field for several years.

**Review Assessment: Checking Correctness Of Derivations And Theory:**

I assessed the sensibility of the derivations and theory.

**Review Assessment: Checking Correctness Of Experiments:**

I assessed the sensibility of the experiments.

**Review Assessment: Thoroughness In Paper Reading:**

I read the paper at least twice and used my best judgement in assessing the paper.

---

> ### Public Comment · ~Mark_Adams5 · 2019-11-08
> **Negative Sampling Description is vague**
>
> Thank you for your submission! A couple of questions:
>
> 1. The paper says "the negatives are uniformly sampled from representations in the same audio signal (z) and/or mini-batch". Could you clarify what this "and/or" means? Considering a batch size of 128 and k=12 steps, do you sample uniformly the 10 negatives out of the 128 * 12 representations? Or do you sample from the 128+11 representations that are the union of the current batch representations at a fixed step and the current audio signal at all steps? Does the method used to select negatives change from one experiment to the other? This point is more than just a detail: as the original CPC paper (van den Oord et al. 2018) showed, the method used to select the negative samples (same-speaker vs. uniform on dataset) had a significant impact on the quality of the produced embeddings.
>
> 2. Do you plan to open source the code supporting the experiments of this paper?

---

> > ### Author Response · Authors · 2019-11-14
> > **Response to negative sampling description**
> >
> > 1. We observed sampling negatives from the same audio signal always provide better results. We will remove “and/or mini-batch” from the line.
> >
> > 2. We will be able to release the pre-trained model after publication.

---

> ### Author Response · Authors · 2019-11-14
> **Response to review #3**
>
> Thank you for the thorough review.
>
> We will include more citations to the ZeroSpeech line of work and contextualize our method in terms of it. However, while both our paper and the ZeroSpeech work learn unsupervised acoustic representations, our semi-supervised evaluation is important since it may conceivably require a qualitatively different kind of acoustic features for optimal performance.

---

### Official Review · AnonReviewer1 · 2019-10-23
**Official Blind Review #1**

**Rating:** 3

**Review:**

This paper proposes an unsupervised method for learning representations of speech signals using contrastive predictive coding.
The authors provide results for the speech recognition task, in which they trained their model on up to 8000 hours of speech. The authors provide results on several English benchmark datasets in addition to four low-resource African language datasets.
The authors compared their method to the traditional signal processing representations and show that the proposed method is superior.

My main concern with this submission is its novelty.
The proposed method was previously explored in [1] and presented similar results. If I understand it correctly, the main novelty in this work is the usage of bi-directional models together with more data. However, it is not clear what made the improvements. Considering the fact that such an approach was suggested recently by [1], a detailed comparison with uni-directional models is needed.
For example, in Table 2, the authors provide results for WSJ dataset, however, with no LM decoding. Can the authors provide experiments of WSJ while using LM similarly to [1]?  Moreover, if the authors wanted to eliminate the effect of LM as they stated in the paper, why not calculating Character Error Rates instead or in addition to Word Error Rates? Again, as done in [1], and in many other papers in the field [2].

Additionally, in Table 1 and Table 5, the error rates seem pretty high, especially for the baseline model, did the authors investigated different architectures/stronger ones for these tasks? Different representations such as LogFilterBanks / MFCCs?

I'm willing to increase my score, in case the authors will address my concerns. However, at the moment, I do not see much novelty in this paper comparing to previous work. Additionally, the authors are missing an essential comparison to previous work so we could better understand the contribution of this paper.

Minor comments: "using a simpler convolutional architecture than is common" -> should be rephrased.


[1] Schneider, Steffen, et al. "wav2vec: Unsupervised Pre-training for Speech Recognition." arXiv preprint arXiv:1904.05862 (2019).

[2] Adi, Yossi, et al. "To Reverse the Gradient or Not: an Empirical Comparison of Adversarial and Multi-task Learning in Speech Recognition." ICASSP, 2019.


**Experience Assessment:**

I have published one or two papers in this area.

**Review Assessment: Checking Correctness Of Derivations And Theory:**

N/A

**Review Assessment: Checking Correctness Of Experiments:**

I carefully checked the experiments.

**Review Assessment: Thoroughness In Paper Reading:**

I read the paper thoroughly.

---

> ### Author Response · Authors · 2019-11-14
> **Response to review #1**
>
> Thank you for the review.
>
> We would like to clarify the novelty of our work and its relation to published work. The representation learning objectives and model architectures are indeed quite similar to [1, 2]; however, we did not intend to imply that either the architecture or training objective is what makes this paper novel, rather we contribute three significant results:
> 1. We demonstrate the feasibility and advantages of pre-training on large-scale and noisy data.
> 2. We demonstrate robustness on out-of-domain evaluation that large-scale pre-training provides.
> 3. We demonstrate that large-scale pre-training results in representations that are universal, as demonstrated by performance on low-resource languages.
> All three of these points are new compared to results shown in previous papers.
>
> Moreover, these are important aspects of representation learning. And, not only do we explore them systematically for the first time, but we also apparently find that large-scale pretraining reverse a trend toward worse performance with larger data that was hinted at in [2], where they show that increasing the amount of pre-training data did not lead to improved downstream performance. Also, in our replication, the models trained only on Librispeech data did not perform well in out-of-domain evaluations or in low-resource languages, demonstrating the importance of diverse kinds of pre-training data - again, a novel and important result. Those findings are of considerable interest since certainly a major benefit of unsupervised learning is being able to improve the robustness of ASR systems, which is a long standing challenge. The low-resource aspect has been investigated in Zerospeech challenge. We will explain the connections in the final version of the paper.
>
> Regarding the effect of the LM on the WSJ task- again, the point of this paper is the robustness of the acoustic representations across datasets, domains, and languages. The results of our in-domain setup demonstrate that these representations are adequate for this setup, but exploring the in-domain setup in detail distracts from the point of the paper.
>
> Regarding the minor tweaks to the model: The improvements to the model (bidirectional context network and dense residual connections) certainly improved the performance to the level that our representations only needs 10% of the labelled data to achieve the same result as the same model trained on spectrogram features using 100% of the training data.
>
> We used exactly the same model architecture, learning rate schedule for different features. We fixed the model to DeepSpeech2 and tuned the learning rate and its scheduler for baseline features (not for our features). It is quite likely we could further improve the results with our features if we re-tuned the hyperparameters, but the scientific point we wished to make was already made.
>
> [1] Representation Learning with Contrastive Predictive Coding, van den Oord et al.
> [2] Wav2vec: unsupervised pre-training for speech recognition, Schneider et al.

---

> > ### Public Comment · ~Michael_Auli1 · 2020-01-22
> > **wav2vec does benefit from more data**
> >
> > >> large-scale pretraining reverse a trend toward worse performance with larger data that was hinted at in [2]
> > This is a misinterpretation of our results. Almost all experiments show that training on more data helps, except for one instance where there was a 0.04 WER degradation on test when adding 8% more data - far from significant. Please fix this in the current draft as it misrepresents our findings.

---

### Official Review · AnonReviewer2 · 2019-10-23
**Official Blind Review #2**

**Rating:** 6

**Review:**

This paper investigates an unsupervised learning approach based on bi-directional contrasive predictive coding (CPC) to learning speech representations.  The speech representations learned using 1k and 8k hours unlabeled data based on CPC are shown to be helpful in semi-supervised learning ASR tasks in terms of sample efficiency, WER and cross-domain robustness. The reported work is interesting and may have value to the speech community.  Regarding the paper, I have the following concerns.

1.  In terms of semi-supervised learning ASR, I think any proposed approach should compare with the "naive" way of doing it. That is, use a high-performance ASR model to decode the unlabeled data and use the decoded pseudo-truth as the ground truth to train an acoustic model with an appropriate capacity.  In my experience,  many of the "novel" approaches can not outperform this "naive" method.  I would like to see this as a baseline for the semi-supervised learning experiments.

2. In sec. 3.1 on the setting of unsupervised learning, the authors state that "all audio signals have a sampling rate of 16KHz". This is obviously not true for the Switchboard data in Table 6 in Appendix A, which has a sampling rate of 8KHz as they are telephony signals.   The authors should clarify.

3.  It is not clear to me why the authors use two different ASR models (DeepSpeech2 small and TDNN). Why not stick to one architecture but adjust the model capacity?

4.   I wonder if the latent features learned by CPC can be complementary to the conventional features such as logmel ? How does it perform if  the two are simply concatenated as the input to the acoustic model?

P.S.  rebuttal read. I will stay with my score.

**Experience Assessment:**

I have published one or two papers in this area.

**Review Assessment: Checking Correctness Of Derivations And Theory:**

N/A

**Review Assessment: Checking Correctness Of Experiments:**

I carefully checked the experiments.

**Review Assessment: Thoroughness In Paper Reading:**

I read the paper at least twice and used my best judgement in assessing the paper.

---

> ### Author Response · Authors · 2019-11-14
> **Response to review #2**
>
> Thank you for the review.
>
> We address review concerns here:
> 1. Our focus in this paper is on completely unsupervised acoustic representations, as well as the properties they confer on the ASR systems that use them. The self-training approach you suggest can indeed be a good way to improve the performance of a system, but it is a different research question that is beyond the scope of this paper. We will identify this as a related strategy.
> 2. The switchboard data was upsampled to 16kHz- we will clarify in the paper.
> 3. We wanted to include TDNN because it is the state of the art in supervised ASR. However, we also wanted to be able to work with a simpler model class than is standard, and DeepSpeech2’s recurrent architecture (which is close to SOTA) meant it could be more easily shrunk without changing the receptive field (which would have happened had we removed layers from TDNN). An alternative would have been to reduce the capacity of the TDNN model without changing the receptive field size, but deep low-capacity layers are difficult to train, and we felt would have led to higher variance results.
> 4. Thanks for the suggestion. We will run these experiments in ongoing work.

---

### Decision · Program_Chairs · 2019-12-19

**Decision:**

Reject

**Comment:**

The paper focuses on learning speech representations with contrastive predictive coding (CPC). As noted by reviewers, (i) novelty is too low (mostly making the model bidirectional) for ICLR (ii) comparison with existing work is missing.